

# Use of electrochemical sensors for measurement of air pollution: correcting interference response and validating measurements

Eben S. Cross[1], David K. Lewis[1,2], Leah R. Williams[1], Gregory R. Magoon[1], Michael L. Kaminsky[3], Douglas R. Worsnop[1] and John T. Jayne[1]

[1] Center for Aerosol and Cloud Chemistry, Aerodyne Research, Inc., Billerica, MA 01821 USA
[2] Department of Chemistry, Connecticut College, New London, CT 06320 USA
[3] Massachusetts Institute of Technology, Cambridge, MA 02139 USA

*Correspondence to*: Eben Cross (escross@aerodyne.com)

**Abstract.** The environments in which we live, work, breathe, and play are subject to enormous variability in air pollutant concentrations. To adequately characterize air quality, measurements must be fast (real-time), scalable, and reliable (with known accuracy, precision, and stability over time). Low-cost AQ sensor technologies offer new opportunities for fast and distributed measurements, but a persistent characterization gap remains when it comes to evaluating sensor performance under realistic environmental sampling conditions. This limits our ability to inform stakeholders about pollution sources and inspire policy makers to address environmental justice air quality issues. In this paper, initial results obtained with a recently developed low-cost air quality sensor system are reported. In this project, data were acquired with the ARISense integrated sensor package over a 4-month time interval during which the sensor system was co-located with a state-operated (Massachusetts, USA) air quality monitoring station equipped with reference instrumentation measuring the same pollutant species. This paper focuses on validating electrochemical sensor measurements of CO, NO, $NO_2$, and $O_3$. Through the use of High Dimensional Model Representation (HDMR), we show that interference effects derived from changing environmental conditions and the ambient-gas concentration mix encountered at an urban neighborhood site can be effectively modelled for the Alphasense CO-B4, NO-B4, NO2-B43F, and Ox-B421 sensors, improving the credibility of air pollutant measurements made with these sensors.

## 1. Introduction

Protecting the air environment is one of the greatest public health challenges, affecting all nations on earth (WHO, 2014). For the past half century, developed countries have made an effort to measure concentrations of major pollutants known to degrade health or damage plants and physical structures. Generally, the focus has been on the most populated areas, with an intent to assess daily, monthly or annual concentrations on a regional basis. While greater spatial and time resolution has been desired, the costs of purchasing and operating instruments sufficiently robust, accurate and free of interferences to generate reliable data has been prohibitive – an instrument to assess a single pollutant at ambient levels can cost many tens to hundreds of thousands of dollars.

In this situation it is therefore easy to understand the motivation to develop inexpensive, rapid-response air quality (AQ) monitoring devices that can be deployed in large numbers around point sources or throughout specific neighborhoods, to create the desired high spatial and temporal resolution AQ data grid (Snyder et al., 2013; Kumar et al., 2015; McKercher et al., 2017). Indeed, within the past decade, researchers, entrepreneurs, and manufacturers have pursued the development, deployment, and evaluation of low-cost devices that measure air pollution (Mead et al., 2013; Williams, 2014b; Masson et al., 2015; Jiao et al., 2016; Lewis et al., 2016; Castell et al., 2017; Mueller et al., 2017).

While electrochemical (EC) sensors have formed the basis for workplace and hazardous leak detection applications for many



decades (Stetter and Li, 2008), their transition from workplace to ambient air is accompanied by much lower target concentration ranges over which the sensors must accurately measure the analyte species of interest (Borrego et al., 2016). Coincident with the need to resolve much lower concentrations is the need to fully understand and model the influence of non-analyte interferences resulting from changing temperature, humidity, pressure, or other gas molecules that may compete with the oxidation/reduction

reactions occurring at the working electrode of a given EC-sensor (Mueller et al., 2017). Unless great care is taken when measuring ambient air pollutants, interferences may result in reported pollutant concentrations that are orders of magnitude greater than the true values. At the core of this quantification challenge is the fact that electrochemical sensors rely on resolving very small changes in current (µA), and in turn, reliably converting that raw sensor signal into a concentration. The path from raw sensor output to concentration requires (1) a mechanical design that provides consistent, empirically validated sampling of

the ambient air, (2) low-noise electrical circuitry (potentiostats) to amplify and resolve small changes in current, (3) electronic filters to remove electrical transients (e.g., radiofrequency (RF) interference) and (4) a method for converting raw signal to concentration that takes into account calibration and interference data.

In order to calibrate and characterize interferences, laboratory and/or field based co-location experiments must be executed spanning the full range of pollutant concentrations and ambient sampling conditions that may be encountered in an actual stand-

alone deployment. Deploying low-cost AQ sensor systems in the absence of such calibration significantly undermines the credibility of the data. Indeed, reports have appeared recently raising concerns about the reliability of data produced from inexpensive monitoring devices containing EC-sensors (Lewis and Edwards, 2016).

This paper describes results obtained from a newly developed, integrated low-cost EC-sensor system, ARISense, which has been developed at Aerodyne Research, Inc. for simultaneous, real-time measurement of a wide range of ambient-level

atmospheric pollutants and accompanying meteorological metrics. We will describe the mechanical and electronic design of the ARISense system, and demonstrate a field-based calibration technique that combines co-located measurements with a High Dimensional Model Representation (HDMR) of the interferences. Our results show that low-cost EC-sensor systems can provide reliable measurements of air pollution under real-world ambient concentrations.

## 2. Experimental

### 2.1 ARISense

The ARISense system used in the present study (called v1) was equipped to measure ambient levels of five gaseous pollutants (CO, NO, $NO_2$, $O_3$, and $CO_2$), atmospheric aerosol particles (0.4 – 17 µm in diameter), and related meteorological and environmental parameters (temperature, T, pressure, P, relative humidity, RH, wind speed/direction, solar irradiance, and noise). Mechanical drawings of the instrumented ARISense system are shown in Figure 1. Each ARISense system is housed in a

NEMA weather-proof enclosure (Polycase, PN: YH-080804; 21.8 cm L × 13 cm D × 21.8 cm H) weighing approximately 2.7 kg fully integrated. ARISense v1 is designed for stationary fixed-site monitoring with access to 120-240V AC power, exterior pole/surface mounting hardware and a consistent sampling orientation relative to the ground.

ARISense v1 contained the following EC-sensors (purchased from Alphasense, Ltd.; UK): Carbon monoxide (CO-B4), nitric oxide (NO-B4), nitrogen dioxide (NO2-B43F), and total oxidants (Ox-B421). (More recent versions of ARISense have been

upgraded to model Ox-B431.) The integrated system also includes a non-dispersive infrared (NDIR) carbon dioxide ($CO_2$) sensor (Alphasense Pyro-IRC-A1) and an optical particle counter (OPC) for measurement of particulate matter size distributions (number-count; ~0.4 ≤ $d_p$ ≤ 17 µm over 16 size bins; Alphasense OPC-N2). The following environmental and meteorological measurements are also included: Relative humidity/temperature sensor (Sensirion AG, PN SHT21), barometric





pressure/temperature sensor (BOSCH, PN BMP180), solar intensity sensor (OSRAM Opto Semiconductors, PN: BPW 34), and a microphone for audible noise detection (CUI, Inc. PN: CMC-5044PF-A). An anemometer (Davis Instruments, Vantage Pro 6410) for wind speed and direction was mounted to the top of the ARISense NEMA enclosure, measuring conditions ~60 cm above the sampling inlets (see Fig. 1 for reference).

ARISense electronics were designed to integrate all sensor measurements into a unified data acquisition framework and provide user access/control over the system's configuration and operation. EC-sensor signals were collected and processed by custom built electronics designed to minimize noise and amplify raw signals (i.e., potentiostat circuitry). Connectivity for v1 systems was enabled via hard-line CAT-5 ethernet connections (Lantronix XPort-Pro). Data was saved at user-defined sampling intervals (5-60s) onto a local USB drive and (if internet-connected) to the ARISense database (https://arisense.io/), where data is

available for real-time visualization and download. Upgraded ARISense systems configured for cellular connectivity and stand-alone solar power are currently under development.

    The ARISense system has two sampling inlets, one for measuring gas-phase pollutants and the other dedicated to the measurement of particulate matter. In both cases, the air flow is driven by small DC-powered fans embedded at the downstream end of the sample flow path, minimizing the loss of sticky or reactive gas molecules ($NO_2$, $O_3$) or particles due to surface

reactivity or deposition. The gas sample flow includes both an intake and an exhaust port in the NEMA enclosure, protected from water penetration via 3D-printed rain hoods (Formlabs; Form 2, Stereolithography 3D printer) mounted to the exterior of the case (see components H, I in Fig. 1). The gas sampling flow manifold and internal PCB mounting brackets were also 3D-printed. Laboratory tests reveal that the 3D-printed material is inert to $NO_2$ and $O_3$ and does not result in significant losses of either species when sampling ambient-level concentrations. The gas sampling manifold provides a consistent, compact interface

for the 4 electrochemical sensors as well as the $CO_2$ sensor. The manifold includes an embedded RH/T sensor positioned adjacent to the electrochemical cells which is used to model the temperature and relative humidity-derived interference effects on the raw sensor response.

    The particle inlet is on the bottom face of the NEMA enclosure (Fig. 1C). Given the body of evidence implicating $PM_{2.5}$ concentrations in adverse health outcomes (Lim et al., 2012), recent years have seen substantial growth in the development,

evaluation, and deployment of low cost OPCs (Holstius et al., 2014; Williams, 2014a; Han et al., 2017; Zikova et al., 2017). The principal measurement challenge of these devices is the minimum size detection limit, often $d_p \geq 0.5$ µm (for devices that cost ~\$250 to \$800) or $d_p \geq 1.0$ µm (cost ~\$15 to \$200). Unfortunately, given these size detection limits, such low-cost OPCs are inadequate when the accumulation mode aerosol size distribution peaks at $d_p \leq 0.25$ µm, which is typical in most urban locations. Low-cost OPC size detection limits also make near-field particulate combustion emission characterization (i.e., near roadways)

very challenging since the combustion mode of particles is typically $d_p < 0.1$ µm. A detailed assessment of the ARISense particulate measurements in laboratory and field experiments will be provided in a subsequent manuscript.

### 2.2 Measurement site

    Two ARISense systems (indicated with yellow circles in Fig. 2) were deployed south of Boston, MA from July to November, 2016. This initial deployment of the ARISense systems was in conjunction with an existing 4-node network (the Dorchester Air

Quality Sensor System (DAQSS) project) established in January of 2016. The DAQSS node locations are indicated with green markers on the map. The neighborhoods of Roxbury and Dorchester are among Boston's largest and most economically diverse, including low-income residential areas interspersed with light and heavy industry, as well as the Interstate 93 corridor which runs along the eastern edge of Dorchester. Given their location and activities therein, Dorchester and Roxbury experience a high frequency of automobile, commercial truck, and heavy duty diesel traffic, much of which is constrained to stop-and-go driving





patterns on congested, narrow streets, in close proximity to housing and pedestrians. Asthma is a serious concern for residents of Roxbury and North Dorchester. From 2001 to 2010 adult resident asthma rates in North Dorchester doubled from 9 to 18%, whereas city-wide asthma rates in 2010 were 11% (Backus, 2012). The original DAQSS deployment and initial ARISense proof-of-concept efforts were motivated by the need to assess the viability of lower-cost AQ sensor systems in communities

suffering from environmental health knowledge gaps. In order to validate our approach, each ARISense system was co-located with a Massachusetts Department of Environmental Protection (MA DEP) air quality monitoring station (indicated with red circles on the map) for the duration of the present study. This paper presents ARISense and MA DEP reference data for the Roxbury site located adjacent to Harrison Avenue in Dudley Square (latitude: +42.3295 longitude: -71.082619). Forthcoming papers will present results from the DAQSS project and the I-93 ARISense node location, covering low-cost AQ sensor results

over longer deployment timescales (12-18 mo.) across multiple types of microenvironments in Roxbury and Dorchester.

### 2.3 Reference data

The MA DEP Roxbury air monitoring site (id: 25-025-0042), established in December, 1998, hosts continuous and semi-continuous gas and particle phase measurements. The reference measurements used in this study include ozone ($O_3$, Teledyne Model T400 Photometric Ozone Analyzer), carbon monoxide (CO, Teledyne Model T300/T300M Carbon Monoxide Analyzer),

and nitrogen oxides (NO, $NO_x$, $NO_2$, Teledyne Model T200 Nitrogen Oxide Analyzer). The reference $NO/NO_2$ measurement is based on chemiluminescence. This method relies on converting NO molecules to $NO_2$ via exposure to $O_3$. Operationally, there are two measurements channels, one for NO alone and one for total $NO_x$. In the NOx channel, a catalytic-reactive converter is used to convert any existing $NO_2$ molecules to NO, prior to exposure to $O_3$. $NO_2$ concentrations are determined by taking the difference between $NO_x$ and NO. Additional on-site reference measurements include a meteorological tower (relative humidity,

temperature, pressure, wind direction, wind speed, solar intensity; MetOne), $PM_{2.5}$ (BAM, Beta Attenuation Mass Monitor), $PM_{10}$, black carbon, and several off-line gravimetric filter samplers including $PM_{2.5}$ speciation. Given its level of instrumentation, the Roxbury location is considered an N-core site within the DEP network of monitoring stations across the state and provides critical data comparisons for determining the viability of low-cost AQ sensor systems. For the current study, DEP provided real-time (1-minute average) pollutant concentration data files from its reference gas analyzers to permit data

comparisons with the ARISense EC-sensor response under rapidly changing conditions of pressure, temperature, humidity, and ambient gas concentrations.

### 2.4 ARISense Calibration

Calibration is a critical issue for trusting the output of EC-sensors. Recent papers (Lewis et al., 2016; Castell et al., 2017) have highlighted that the lack of rigorous calibration protocols for low-cost AQ sensors results in significant potential error when

the sensor system is deployed in ambient conditions. For example, Mead et al. (2013) modelled the temperature dependent baseline drift of an Alphasense NO sensor using an exponential curve fit through 24 hours of ambient data. Their analysis revealed that temperature-derived baseline-drift could exceed a +600 ppb bias, if unaccounted for in their calibration (sampling between 20 and 28 C). Considering that the ambient NO concentration range encountered in the current study was 0 - 200 ppb with temperatures varying from 5 – 45 C (5-min averages), modelling the NO-B4 sensor temperature-derived interference is

crucial to obtaining useful measurements from the sensor. As Mead et al. (2013) point out, when measuring gas concentrations at the ppb level, temperature and humidity interference effects have a first-order impact on quantification, whereas drift in sensitivity over time has second-order effects (much smaller in magnitude than temperature or humidity influence). Both first



and second order effects need to be correctly parameterized in order to apply low-cost sensors to ambient outdoor air quality measurements.

Alphasense provides some guidance to customers regarding calibration and temperature-compensation of electrochemical sensor response (Alphasense Application Note #AAN 803-03, December 2014). This document highlights the utility of including a fourth electrode in their B4-series electrochemical sensors such as were used in this study. The purpose of this fourth electrode (called the auxiliary electrode, AUX) is to provide a real-time correction for environmentally-derived interferences at the working electrode (WE). The AUX electrode is comprised of an identical catalyst to that of the WE and is designed to mimic the WE's response to environmental changes such as temperature, pressure, and humidity. Since the AUX electrode is fully submerged in the electrolyte and directly below the WE, the AUX signal should be blind to the target analyte gas species which readily oxidize or reduce at the WE surface (which is exposed to the air on one side and the electrolyte layer on the other). In an ideal world, a simple subtraction of the current generated at the AUX electrode from the current generated by the WE would provide a signal that is linearly proportional to the target analyte over the full concentration range of interest. Unfortunately, we have found that in practice the AUX electrodes in most sensors are not able to track the changes in the corresponding WE over the nominal operational temperatures of the system. Specifically, at temperatures > 25 C, the AUX electrode response lags that of the working electrode and in some cases (CO-B4, for example), the WE and AUX electrode currents diverge as temperature increases (i.e., the WE current decreases with increasing temperature while the AUX electrode current increases with increasing temperature). In this case, recording just the differential current without correction leads to an increasingly negative concentration error for CO at temperatures above 25 C. Alphasense provides users with a table in which, for each sensor model, the user can identify a correction constant to use to compensate for observed behavior at specified temperature ranges. At temperatures ≤ 20 C the Alphasense documentation shows that differential measurements remain fairly stable in comparison to the higher temperature conditions. The Alphasense approach to temperature compensation also requires the use of four static constants for each individual EC-sensor – subtracting specific electronic and zero currents from both the WE and AUX electrodes, prior to calculating the difference. While there are some advantages to the additional information provided by the AUX electrode, at temperatures higher than 25 C, the disparate response between the two electrodes can complicate quantification steps considerably.

In practice, we have found that the manufacturer's recommended WE and AUX electrode corrections do not lead to pollutant concentration values of acceptable accuracy for ambient air analysis. In addition, the EC-sensor response is impacted by other environmental conditions besides temperature, such as relative humidity and the concentrations of other species. At the low concentrations present in the atmosphere (10s-100s ppb) characterizing the full interference response is critical to achieving reliable measurements. In this work we demonstrate the use of a multi-dimensional mathematical modelling approach (HDMR) that has the ability to adequately identify and quantify the complex EC-sensor response to multiple environmental variables and interfering gas species simultaneously.

### 2.5 High Dimensional Model Representation (HDMR)

The ARISense system uses HDMR to convert the raw sensor outputs into units of concentration in parts-per-billion by volume, ppb. HDMR is a numerical method consisting of a general set of quantitative model assessments and analyses for capturing input-output system behavior without reliance on a physics-based model or the sensor manufacturer's empirical correction procedure. When applied to a set of experimental data (with sufficient variability), it can produce a mathematical model relating user-defined input variables to output variables of interest; the resulting model can capture the intricate interdependencies of the variables and provide a mathematical description of the system that is otherwise difficult or impossible





to describe with a physics-based model. The HDMR model can be used to identify and quantify which variables and variable interactions have the most impact on the data reduction. Aerodyne has implemented the HDMR method in a software tool called *ExploreHD*, providing graphical and command line user interfaces to HDMR algorithms.

The details of the HDMR algorithms used here are discussed in detail elsewhere (Li and Rabitz, 2010; Li et al., 2010; Li and

Rabitz, 2012; Li et al., 2012). One of the key underlying tenets of the HDMR framework is that many input-output relationships for complex physical systems can be captured adequately by the correct combinations of input variables, even in systems with high-dimensionality in input variables. Each component function provides an additive contribution to the overall model prediction. The modelling process involves three steps. In the first step, the user specifies a maximum variable interaction order (for example an interaction order = 2 would allow any two variables to interact), and the HDMR algorithm considers orthogonal

component functions (in this case, cubic polynomials) involving all possible variable combinations up to the maximum specified order. In the second step, a statistical analysis is performed to identify the input variables and combinations of input variables that contribute significantly to changes in the output of interest. In the final step, coefficients for component basis functions are calculated through a least squares analysis that minimizes the deviation between HDMR model prediction and the training data. The coefficients and the associated orthogonal basis functions determined through the above analysis together define an HDMR

model for the input-output relationship under consideration.

In the current study, the HDMR approach uses the raw EC-sensor output and environmental variables to model the multi-dimensional surface between sensor output and the reference concentration. We used approximately 35% of the dataset to train the model. Sensor interference can be a product of the combined influences of temperature, humidity, pressure, non-analyte gas species, etc. The structure of the computational model accounts for both absolute (i.e., highest to lowest concentrations) and

transient ($\Delta x/\Delta t$) changes in the sampling conditions encountered by the sensor system. By spanning three seasons in the Northeastern United States, a wide range of environmental conditions was captured within the training window for the model. This emphasizes the advantage (i.e., variability in sampling conditions) and disadvantage (extended time-span) of a field-based co-location approach to sensor calibration.

It is important to recognize that the data presented in this paper were recorded over a 4-month sampling interval (July 7,

2016- November 23, 2016). All four electrochemical sensors used in this study were first removed from their packaging on May 9, 2016. That means that from out-of-package, the sensors had aged ~6.5 months by November 23. For the purposes of this study, we are assuming that the sensitivity of each of the electrochemical sensors did not appreciably drift over this time interval. This means that the HDMR model built in this case does not take sensor age into account. In subsequent studies we will analyze sensor response over longer deployment timescales to investigate the importance of including a time-based model parameter to

track and correct for drift in sensor response with time.

## 3. Results and Discussion

### 3.1 ARISense meteorological/environmental data

Continuous 5-min average non-pollutant data acquired with the ARISense system is shown in Fig S1 of the supplemental, tracking ambient variability in temperature, pressure, humidity, solar intensity, ambient noise, wind speed, and wind direction at

the Roxbury DEP monitoring site. The total sampling timespan covers the transition from mid-summer through late fall in the Northeastern United States (July through November), with meteorological conditions changing from warmer and more humid to cooler and less humid. The ARISense system ran continuously throughout the sampling interval with the exception of a ~ 1-week period during which the node was physically removed from the site for a separate experiment. The directionality of the





wind fields at this site is predominantly from the N to NW (red-maroon) with occasional NE flow (blue-purple). Temperature and humidity measurements shown reflect the conditions within the gas-sampling flow-cell of the integrated system, characterizing the precise environmental conditions at the surface of the electrochemical sensors. Such environmental measurements are critically important for reconciling the interference effects of ambient conditions, especially humidity (water concentration) and temperature, on the raw signal from each electrochemical cell.

### 3.2 ARISense electrochemical sensor data

Time-series of 5-min average raw differential signal ($\Delta$mV = working electrode - auxiliary electrode) obtained from each electrochemical sensor (CO, NO, $NO_2$, and $O_x$) in the ARISense system are shown in Fig. 3. The signal is displayed as a voltage which is linearly related to the current generated within the electrochemical cell. The time-series of co-located reference concentrations are plotted on the right axes for each gas-phase pollutant of interest. The correlations between the raw EC-sensor output and the reference measurements are shown in panels a-d of Fig. 4, with each data point colored by flow-cell temperature.

The raw differential signals obtained from the CO-B4 sensor correlate reasonably well with the CO concentrations measured by the co-located DEP monitor (Fig. 4a), demonstrating the relatively small influence of ambient temperature, humidity or other chemical species on this EC-sensor. The NO sensor also correlates reasonably well with the reference measurements, (Fig. 4b) except at temperatures over 25 C when the EC-sensor overestimates NO by a factor of 2 to 3 compared to lower temperatures. This suggests that the temperature dependence of the working and auxiliary electrodes in this NO sensor do not track one another at sample temperatures > 25 C, and that additional temperature-correction is necessary to obtain reasonable NO concentrations from raw sensor outputs alone.

The $NO_2$ and $O_3$ sensor outputs correlate less well with the reference measurements. The differential NO2-B43F sensor response (Fig. 4c) indicates a strong temperature dependence that is not compensated for by the auxiliary electrode, suggesting that additional temperature compensation algorithms could improve the result. The differential signal from the Ox-B421 electrode shows poor correlation with the reference data overall (Fig. 4d). There is some temperature-dependence, but the additional residual variation suggests that other factors play an important role. The Ox-B421 sensor is comprised of the same catalyst (working and auxiliary electrode material) as the NO2-B43F, and is therefore-sensitive to $NO_2$ in addition to $O_3$. The key difference between these two sensors is the presence of an $O_3$-scrubbing filter upstream of the working electrode in the NO2-B43F sensor package. Laboratory results indicate that the Ox-B421 sensor is ~2x more sensitive to $NO_2$ than to $O_3$ molecules.

In order to demonstrate the varying effect of temperature on the different electrodes, Figure 5 shows raw 2-min average output currents for both the working (darker hue) and auxiliary (lighter hue) electrodes for each electrochemical sensor over a 48-hour time interval. Reference concentrations of each pollutant species are plotted (as red dashed-lines) on the right-hand axes for comparison, and the flow-cell temperature is displayed as a filled histogram in the background of the time-series. Temperature changes in excess of 22 C are observed in as little as 10 hours. Offsets are observed between the working and auxiliary electrodes of all four sensors, but the extent to which the working/auxiliary pair track one another as environmental conditions change is highly sensor-dependent. The CO-B4 sensor appears to be fairly insensitive to the temperature changes encountered, with a positive-going differential reflecting changes in the ambient concentration of CO. In contrast to the muted temperature response of the CO sensor, the NO-B4 sensor shows a strong temperature dependence in both the working and auxiliary signal, with some evidence of lagging temperature response in the auxiliary channel at the highest temperatures encountered. The NO2-B43F shows a clear temperature dependence in the working electrode, but the auxiliary electrode shows less of a response. However, in a few instances in the 48-hour window displayed in the figure, it appears that the NO2-B43F auxiliary response to rapid decreases in temperature (coincident with sunset) is opposite the trend observed for the working





electrode. The Ox-B421 sensor exhibits a temperature-dependent response that largely tracks between the auxiliary and working electrodes.

As Figure 5 shows, the magnitude of the interference signal derived from temperature alone (for NO, $NO_2$, and $O_x$), can easily mask real pollutant-derived variation. The raw signal behavior observed for each sensor type is unique, underscoring the

necessity of species-specific HDMR models to reconcile each sensor type's characteristic interferences. In addition, substantial (~ 2-3x) differences (in sensitivity and baseline) exist for batches of nominally identical sensors measuring the same concentration. Therefore, the HDMR models built for a given integrated system are specific to a given set of sensors, and must be generated for each system separately to achieve reliable concentration data. Within the framework of an individual ARISense system, 4 distinct HDMR models are built, one for each EC-derived pollutant species of interest.

**3.3 HDMR Analysis**

The next step in the analysis was to determine whether statistically significant correlations between raw EC-sensor outputs and reference pollutant concentrations could be extracted from the data shown in Fig. 3 plus the environmental data in Fig. S1. The time intervals over which the model input matrix was generated were chosen to provide comprehensive coverage of environmental variability spanning the July-November sampling interval. It was important to include (1) sensor responses to the

range of gas concentrations encountered in ambient air (near-zero to high concentration transient spikes in pollution), (2) the range of temperatures and various rates-of-change in temperature, and (3) the range of measured water content of the sample air in the flow-cell. The goal was to include a wide enough range of training data to avoid extrapolation errors when applying the model to the rest of the dataset. In total 35% of the full time series data were used in to generate the model (and are indicated with grey bars in Fig. 6). For each sensor, the input data matrix was optimized by systematically testing all possible input

parameter combinations and then evaluating the resultant model's performance against the full co-location dataset. An example of two different input matrices for determining $NO_2$ concentrations is provided in the Supplemental Material (see Fig. S2).

Correlation plots of model-derived pollutant concentrations and reference concentrations for the training data are shown in the middle panels (e- h) of Fig. 4. The 1:1 lines are shown for all species (black dashed lines), and all data points are colored by flow-cell temperature. The lack of a temperature-dependent rainbow in the scatter plots shown in Fig. 4e-h (with the exception of

$O_3$, for which ambient concentrations are expected to be quite temperature-dependent) indicates that we have been able to effectively compensate for the variable temperature dependent response of the different electrochemical sensors, and the WE and AUX electrodes within each cell. The remaining scatter in the correlation plots is random noise attributed to the electrodes themselves and the electronics, plus some inherent scatter in the reference data. The high correlation coefficients ($R^2$=0.76-0.96) indicate that, when trained appropriately, the HDMR data reduction provides dramatically improved compensation for the

interferences that complicate interpretation of raw EC-sensor outputs.

Figure 6 shows the time-series plots for the full dataset analyzed with the HDMR models trained on the (35%) subset of ambient data. Data shown are 5-min averages of the modelled (sensor) and reference gas concentrations. Correlation plots of sensor-derived and reference concentrations for the entire dataset are shown in Fig. 4i-l. The robustness of the linear correlation plots indicates the strength of the model at capturing the ambient variability in pollutant concentrations encountered at the site,

despite wide variations in ambient temperature and humidity over the changing seasons. The higher scatter in the $O_3$ correlation plot can be largely attributed to the fact that $O_3$ is obtained by training the Ox-B421 sensor output to reference $O_3$; with the 2:1 sensitivity ratio for $NO_2$ vs. $O_3$ of the Ox-B421, the variability in ambient $NO_2$ concentrations adds considerable noise to the Ox-B421 sensor signal. The input matrix for the Ox-B421 HDMR model includes the raw data captured with the NO2-B43F sensor, but the inclusion of this additional information only marginally improves the reduction of the Ox-B421 data to $O_3$ concentration.



It should be noted that the Ox-B421 sensor is not the latest version released by Alphasense and improvements may be possible with the design in their most recent model (Ox-B431). This fact highlights the iterative and rapidly evolving nature of sensor components. Low-cost air quality sensor quantification will likely improve over the coming years through advances at the manufacturer level (e.g., Alphasense Ltd., improving materials chemistry/catalyst and sensor-design) and system calibration
level (e.g., Aerodyne Research, Inc., further developing ARISense HDMR interference modelling).

While Figures 4 and 6 illustrate that the system is capable of determining valid gas phase concentrations across a wide range of environmental variability in temperature, RH, and absolute concentrations, it does not speak to the longer-term stability of the sensors (e.g., how much does the baseline and sensitivity of each electrochemical sensor change with time). However, it should be noted that sensor aging cannot have had a major impact on the data reported here or it would have been impossible for the
HDMR model to converge this well without entering electrode age as one of the variables. In the actual analysis, each data point for each variable had equal weight, whether it was at the beginning, middle or end of the time span. It is to be expected that aging of EC-sensors will change their sensitivities, due to electrolyte evaporation or dilution, entrapment of contaminants, and repeated exposure to wide swings in T or RH. It will be important to establish the time span over which a given set of EC-sensors and the HDMR training of that sensor set can be expected to return reliable pollutant concentration values, using a longer
duration empirical data set; such a study is in progress.

## 4. Conclusion

This study demonstrates that low-cost air quality sensor systems can adequately characterize pollution concentrations and have the potential to add a highly resolved local AQ data-layer to existing pollutant monitoring infrastructure. The ARISense system is a first step toward understanding the extent to which quantification efforts can yield useful results from such systems.
Referring back to the map displayed in Figure 2, it is striking to consider that only 4 official monitoring stations exist within the Boston-metro area (pop. ~700,000). With regard to the Roxbury DEP site 1-minute average reference data, it is important to note that 1-min data files are not typically reported or accessible from regional air quality monitoring sites. Instead, pollutant concentrations are usually reported on 1, 8, or 24-hour averages in accordance with the operational constraints of the measurement device and relevant air quality regulations being enforced. This sampling paradigm is consistent with the regional
focus of federal and state monitoring goals, and financial constraints imposed due to the expense of the instrumentation and operating costs of a given AQ monitoring station. But as one considers the pollutant sources that can impact their local area, disproportionate pollution impacts emerge in some neighborhoods more than others. Across urban landscapes, air pollution is inherently heterogeneous, subject to sharp concentration gradients over fast (sub-minute) and short (100's of meters) scales. In order to establish a more rigorous assessment of such disparate impacts, distributed sensor networks are needed to achieve high
enough spatial resolution to inform intra-neighborhood differences in air quality. Through such advances, researchers, regulators, and community members can improve their understanding of the pollutant sources that have a disproportionate influence on local AQ. As sensor technologies (and calibration/modelling efforts) continue to improve, the local AQ data layer could play a key a role toward empowering environmental justice advocates to initiate change.

It cannot be overstated that EC-sensor systems such as ARISense can return reliable data only if calibrated over the full range
of pollutant concentrations and meteorological parameters that will be encountered when they are deployed. In the present study, co-location of the ARISense system with the MA DEP reference monitors, coupled with variability of natural and anthropogenic activities, supplied the necessary range of calibration events over the four-month span of the study. In the future, we expect to compress that training period into one week, using a controlled-environment laboratory chamber and mixes of calibration gases





representative of the pollutants encountered under ambient conditions. This compression of the training period is especially important when addressing the challenges of sensor-to-sensor variability, finite (< 36 mo.) sensor lifetime, and premature damage or failures that will require rapid replacement/re-training of integrated systems.

**Acknowledgements**

The authors would like to thank the staff at the Massachusetts Department of Environmental Protection for their support, including access to the DEP monitoring stations and raw 1-min data from the reference gas analyzers used in the current study. We are especially grateful to John Lane, Leslie Collyer, Emmy Andersen, and Thomas McGrath at MA DEP. ESC also thanks Stephen Prescott for his help with the ARISense mechanical-electrical assembly, Xavier Cabral for his contributions to the electrical design, and Conor Mackinson and Wade Robinson for their mechanical design contributions to the ARISense project.

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

20





**Figures**

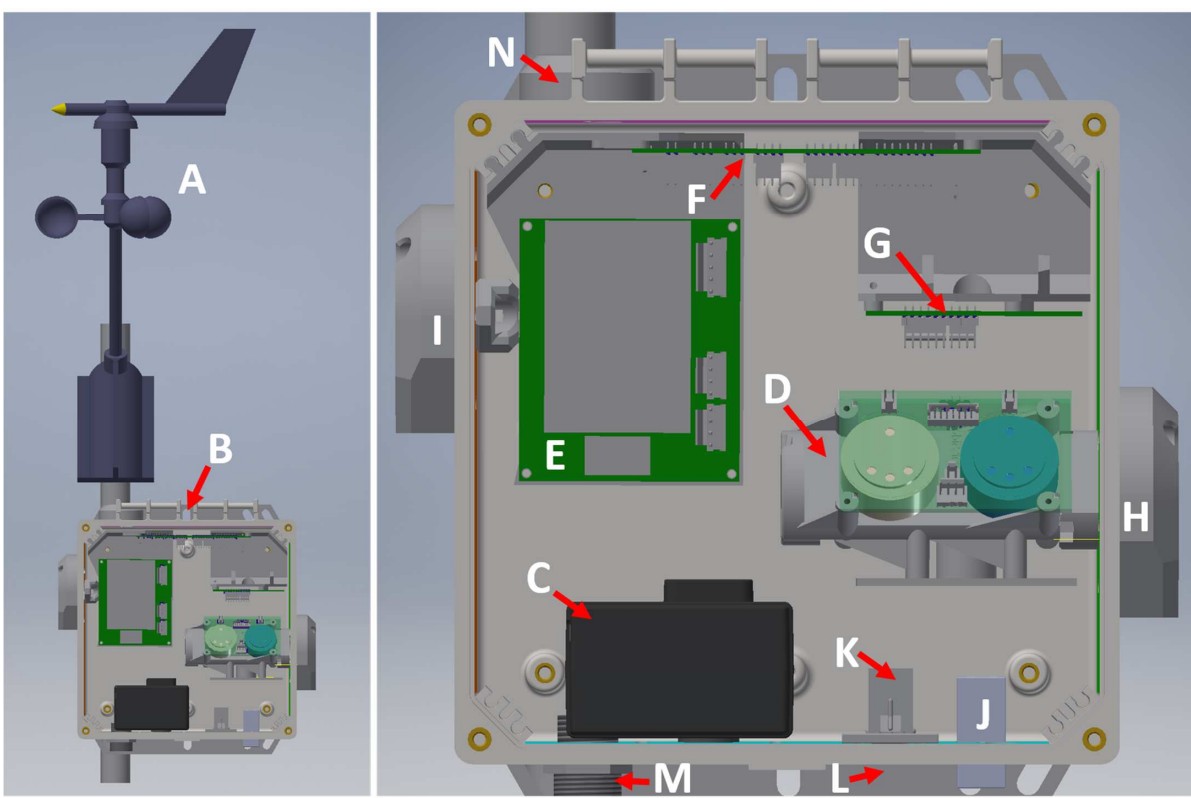

5    Figure 1. Mechanical drawings (wires excluded) showing the main components of the ARISense system. Each system includes an anemometer ('A') mounted to the back-bracket of the NEMA enclosure providing a description of the wind-fields in the immediate proximity to the gas and particle sampling inlets of the system. Mounting brackets for wall or pole-mount configurations attach at position B. Expanded view of the internal components reveals the Optical Particle Counter (C), gas sampling manifold (D) with embedded electrochemical and NDIR and RH/T sensors, transformer/power PCB (E), main controller PCB (mounted vertically within the enclosure)
10    (F), communication PCB for ethernet connectivity (G), gas sampling inlet and exhaust rain hoods (H, I), RJ11 and RJ45 connections for anemometer data and CAT-5 connectivity (J, K), microphone assembly (L), weather tight AC power input (M), and solar sensor assembly for light intensity measurement (N). 3D-printed parts include the gas sampling manifold, rain hoods, exhaust and microphone mounting bracket, solar sensor interface, and PCB mounting scaffold. As described in the text, the gas and particle sampling inlets are decoupled, with the OPC-N2 sampling through the bottom-face of the enclosure to protect from liquid water penetration.

20





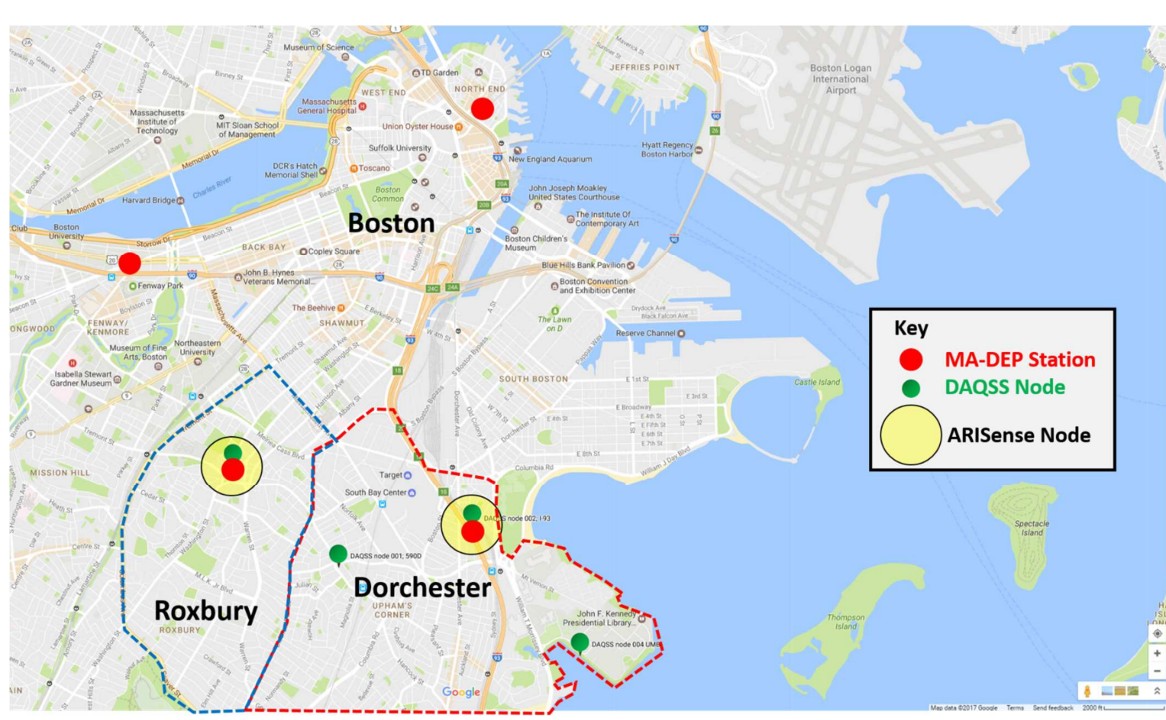

Figure 2. Deployment map showing the locations of the two ARISense systems (yellow circles) and the four metro-Boston DEP monitoring stations (each marked with a red circle). The two ARISense systems were co-located with reference stations at the Harrison Avenue site (in Dudley Square, Roxbury) and Von Hillern Ave. site (~35' off of I-93 North) in Dorchester. The data presented in this manuscript were obtained from the Dudley Square location, an urban neighborhood site, primarily impacted by local combustion sources operating on secondary routes in close proximity to the area. Green markers are shown to indicate the positions of 4 additional sensor nodes deployed as part of the Dorchester Air Quality Sensor System (DAQSS) project, which pre-dated the development of ARISense.

20





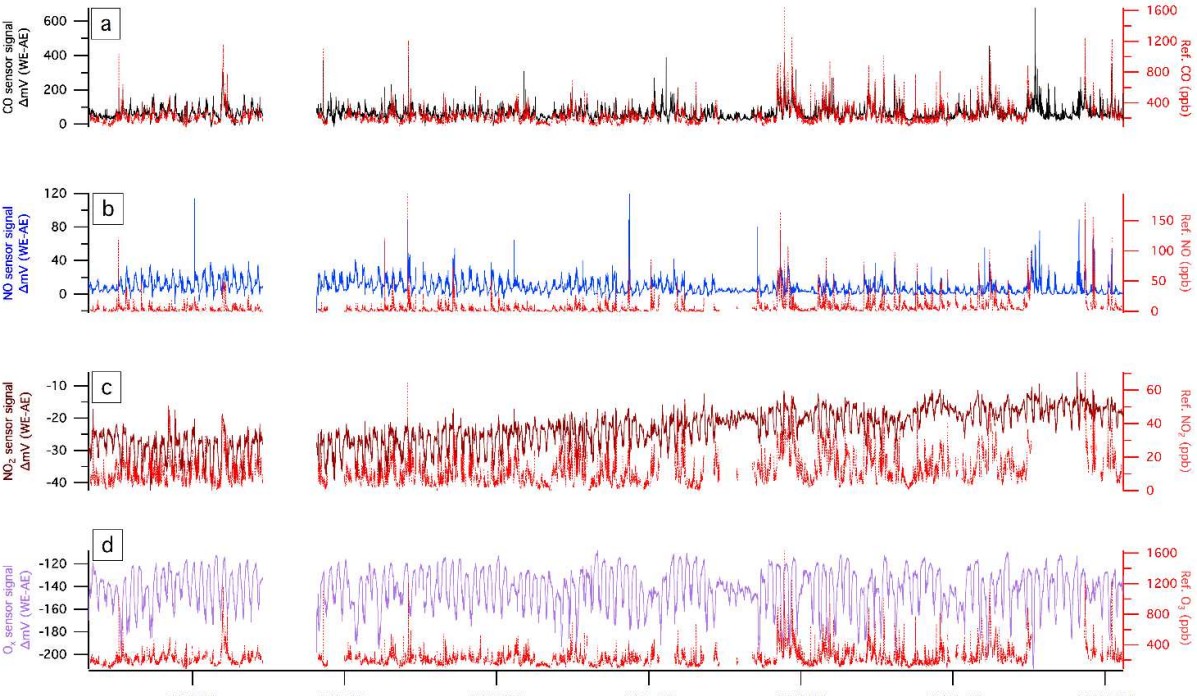

Figure 3. Time-series of 5-min average raw differential voltage (working electrode-auxiliary electrode) obtained from each electrochemical sensor (CO, NO, NO₂, Oₓ). Overlapping time series of co-located reference concentrations (in red), measured with Federal Equivalent Method (FEM) monitors from the Roxbury DEP site (averaged onto the same 5-min time base) are plotted on the right axes for each gas-phase pollutant of interest. In the case of the Oₓ sensor, the ozone (O₃) reference concentrations are plotted.





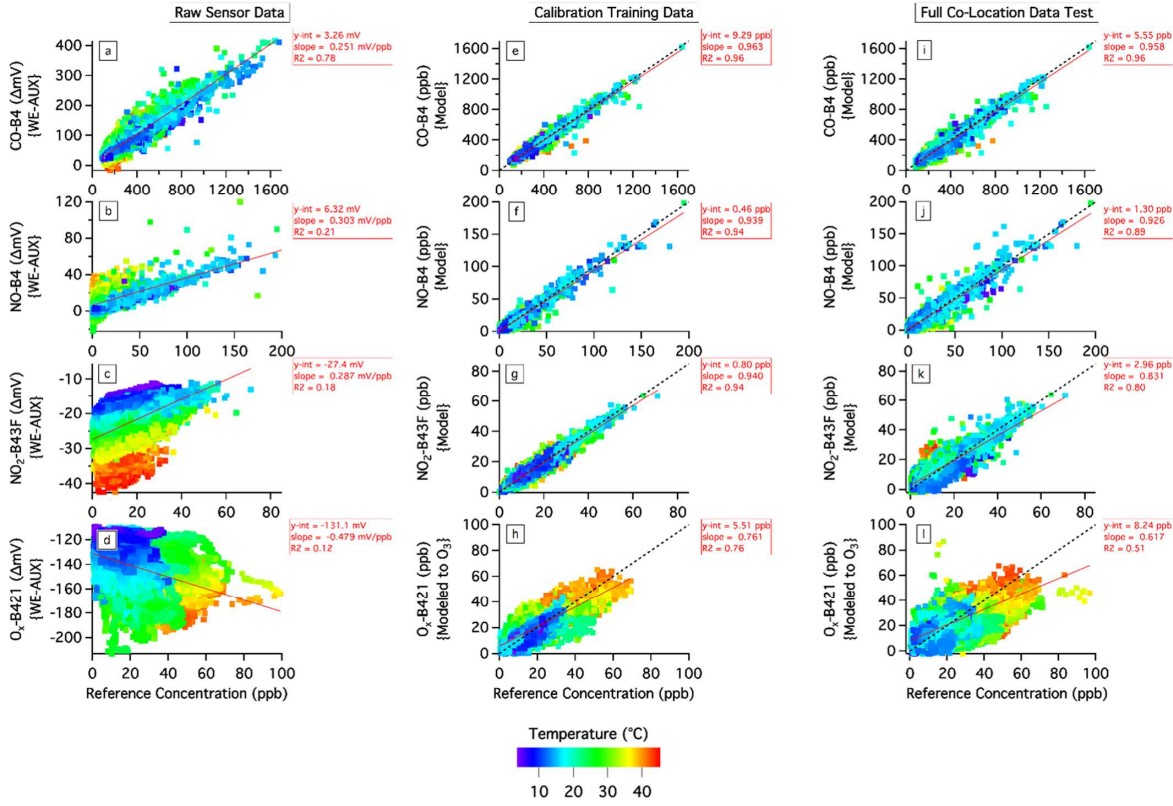

Figure 4. Correlation plots for all electrochemical sensors for (1) raw sensor differential voltage signals (a to d), (2) model output concentrations for the calibration training interval only (e to h), and (3) model output concentrations for the full co-location time-series (I to l). All data shown are 5-minute average values with each data point colored by flow-cell temperature. Linear coefficients are displayed in each panel with the linear fit-line drawn in red. To facilitate evaluation of the model output correlations, a 1:1 line is drawn (black dashed line).





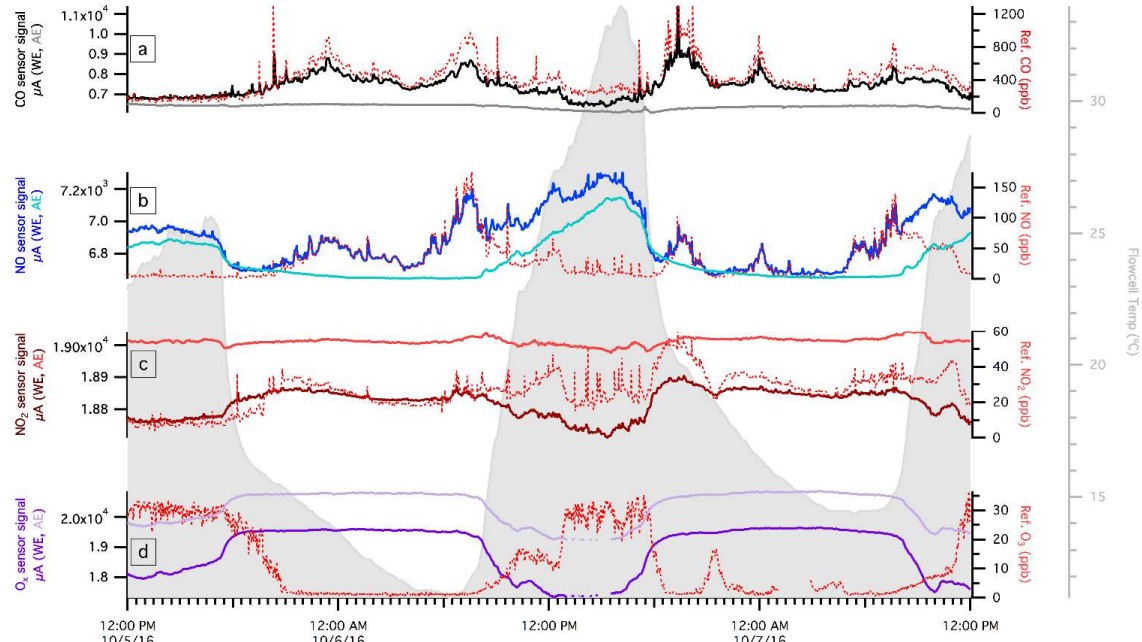

Figure 5. Raw 2-min average signal (in µA) for the working (darker hue) and auxiliary (lighter hue) electrodes for each electrochemical sensor over a 48-hour time interval. Reference concentrations of each pollutant species are plotted (as red dashed-lines) on the right-hand axes for comparison. To illustrate the influence of temperature on the raw electrochemical sensor signals, the flow-cell temperature is displayed as a filled (grey) histogram in the background of the time-series.



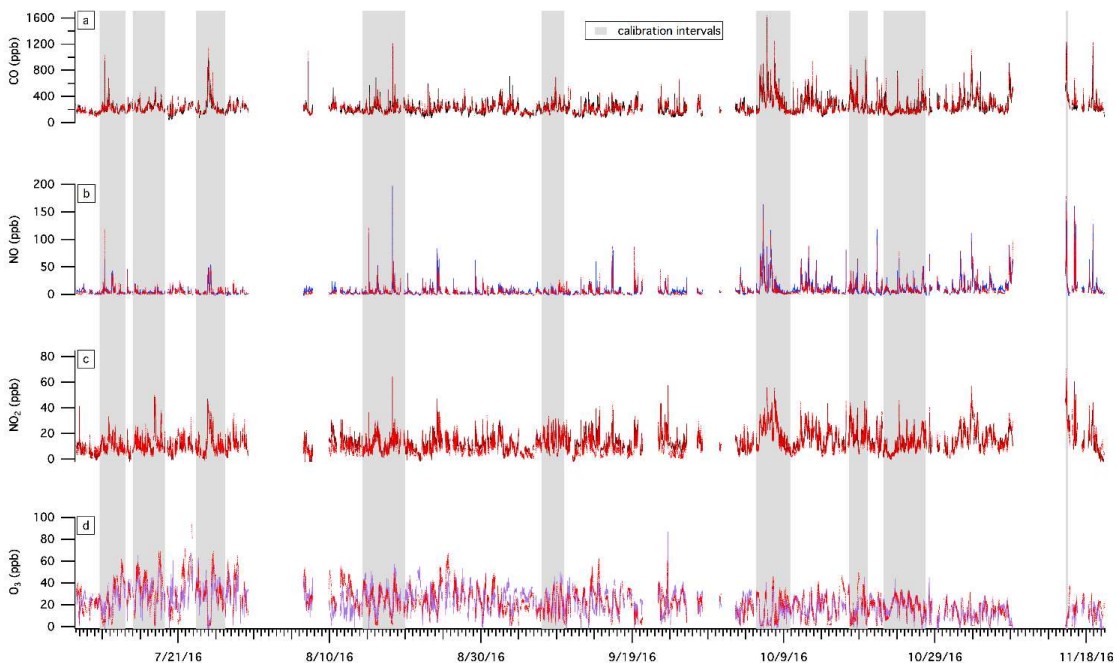

Figure 6. Time-series data illustrating the full dataset analysed with the HDMR model trained on 35% of the data. Data shown are 5-min averages, time-series of the modelled (sensor) and reference gas concentrations. Grey shaded areas indicate time periods over which the model was trained (i.e. the model calibration interval). A unique set of input parameters was used to train each of the different electrochemical cells. In each case, the input data matrix was evaluated and optimized by testing the model across the full co-location dataset. In total 35% of the full time series data were used to generate the model, whereas 65% of the data comprise the true 'test' of model robustness against the ambient variability of environmental conditions and gas concentrations encountered at the site.

