# Peer review of "Use of electrochemical sensors for measurement of air pollution: correcting interference response and validating measurements"

_Atmospheric Measurement Techniques, 2017_

## Referee Comment (RC1) · Anonymous Referee #1 · 30 May 2017

This is a rigorous attempt to illustrate the challenges and utility of deploying 'low cost' air quality sensors in a community. This is a field of growing interest – likely to become more crowded – with important implications across most spheres of atmospheric research. A particularly strength of this work is stressed by the authors in warranting caution in interpretation of data from these types of sensors.

Specific comments: P1, Line 14: perhaps this is better phrased as '. . .address environmental justice issues related to air quality.' P1, Line 22: 'Protecting the air environment. . .' isn't really one of the most important PH challenges. Rather, it is protecting populations from degraded air quality exposure that is important. P1, Line 29: to presume the authors mean US dollars? P8, Line 18: extra word 'in'

General Comments: One issue that is not discussed is the potential for lot variability in sensor performance within a single manufacturer. Whilst the authors provide adequate detail on which make/model has been chosen (P2, L33-40), do these EC sensors exhibit differences within a manufacturer production lot? Or are there differences across different lots?

The paper begins with a discussion on environmental justice (abstract), and includes very specific references to asthma rates in the sampled community (P4, lines 1-4). Context is, of course, important, but these facts seem out of place in this manuscript which is mainly a focus on the technical details of using and interpreting EC sensors.

It was surprising to see a reported temperature range of 5-45 degrees in a northern US city, but the authors later state that this was internal box temperature to assess electrode function under actual operating conditions. When comparing these data, was a correction to ambient temp and RH taken into account (e.g. temp/RH measured by a nearby met station)? For example, if ambient temp and RH were 25 deg and 50%, but the internal temp where 35 deg and 15% because of strong sunlight, one would expect a significant effect, given the apparent sensitivity.

Why was there no data included or discussed for particulate matter or CO2?

In a number of cases, the authors refer to this sensor package as a 'low-cost' replacement for measuring air quality, which could play a key role towards empowering environmental justice (P9, Line 34). The authors are correct in asserting that lower cost sensors likely have a role in improving granularity in air quality monitoring networks, especially in locations with disproportionate air quality burdens, like the relatively low income communities in which this study takes place. But the idea of 'low cost' is a fairly subjective statement that seems meant to broaden the appeal of these products to communities in need. The development of low or lower cost sensor units with an eye towards reducing injustices is a noble and important direction for air quality scientists, but it might provide value to compare this instrument against the few other existing

low/lower cost sensing units that are found in the literature – both in terms of sensor performance and relative cost.

The largest issue seems to be in interpretation and setup of the HDMR model to adjust sensor data to real values. Specifically, the authors state that the model can 'capture the intricate interdependencies of the variables. . .' in order to correct the data and provide guidance to researchers which variables are most impactful (P6, Line 1). These statements presume that the researchers enter in all possible variables that are likely to play a role in sensor performance. Given the relative few number of variables measured, and presumably computed, how can a researcher have confidence that they are accounting for all – or at least most – of the likely variants that may affect their results? The concern here is that there may be other plausible covariates that affect sensor performance. For example, one might imagine a measure of $CO_2$ by NDIR could be affected by water vapor (which is imputed by this sensor package), but also by other ambient IR-absorbing components (that are not measured)?

The authors also note (P 6, Line 8-9) that in the first step of modeling, a user can choose how many variables are selected to interact with one another. How does one quantitatively make this determination?

It is very difficult to discern useful results from Figure 3. Further, we must presume that these data have been validated by the investigators. If so, it is surprising to see spikes of ozone exceeding 1000ppb with some regularity in this location, as observed by the reference monitor.

Figure 5 is a fairly useful illustrative figure that clearly identifies sensor limitations. But it is troubling to see divergence between the EC sensor and the reference sensor in periods of relative stability in temperature. This seems to need further explanation – how does your data compare for this specific time series after it has been modeled?

The authors included a number of variables to consider in adjusting or training the model, but specifically excluded sensor age, noting that the sensors were approximately 6-7 months old at the end of the study and, therefore, should have limited effect on model performance. Firstly, wouldn't it be more appropriate to compare sensor age to manufacturing date, rather than when a package is opened? And second, it is unsatisfying to ignore sensor age as a relevant variable, given the relatively short lifetimes of these sensors. For example, the NDIR lamp and electrode has a lifetime of 2000-6000 hours (according to the manufacturer), depending on lamp light time and the presence of heavy contaminating pollution. This is 80- 250 days, which is not much longer than the study length presented here, and suggests that long term drive is something that should not be ignored.
* * *

---

## Referee Comment (RC2) · Anonymous Referee #2 · 7 Jul 2017

General comments: This paper is timely in describing how to improve the performance of a set of Alphasense electrochemical sensors, which are being widely incorporated into may emerging multipollutant air quality sensor technologies. The paper goes into great depth in exploring causes of sensor measurement artifacts and demonstrates an approach to improve the data quality. However, this paper will have a limited impact if several important issues are not addressed. A recommendation of major changes is suggested, focusing upon these areas of improvement:

1. How are authors defining "good enough" for sensor data quality? They indicate a goal of having credible data and "acceptable accuracy" (line 27), but need to clarify

what they consider to be their target (accuracy, measurement range, etc.) and for what purpose. 2. The authors note in their concluding sentence that "This compression of the training period is especially important..." Currently, they used 35% of a 4 month period of data to develop a complex model to improve the data. Why 35%? What is the performance if only 10% of the data were used? What if only the first week of data were used? Authors have sufficient data to explore the implications of different training periods that would provide important insight to researchers looking to employ sensors and develop study plans yielding reasonable data quality. It is recommended that authors go into substantially more depth to investigate the training period required. 3. Authors should investigate an aging effect – they indicate they will only explore this later, but should at minimum demonstrate whether there is any relationship with the number of "out of box" or "in use" days. In Jiao et al (2016, https://doi.org/10.5194/amt-9-5281-2016), aging was clearly demonstrated in a number of sensor types that incorporate Alphasense sensors. 4. How variable is the performance between identical sensors? How variable are the HDMR models from one RAMP to another? 5. The HDMR analysis is fairly opaque – authors cite papers that describe the approach, but do not provide sufficient detail for this to be reproducible. It is recommended that authors provide more specific information on the HDMR analysis and resulting model in the supplemental information. Given some sensor applications involve real-time transmission and display of data to the public, does the HDMR approach support this or must it be performed post hoc?

Minor comments: 7. Quality of the text on figures needs improvement – recommend not using red font text and ensuring clear, readable axes. 8. Authors compare against DEP monitors – they should indicate what are the detection limits of the monitors and implications for their calibration. Since regulatory monitoring stations are employed to evaluate air quality relative to the NAAQS, detection limits can be an issue in low concentration areas (e.g., some CO monitors have ~300 ppb detection limits, which may be fine for the NAAQS at a ppm level but may be an issue for co-location and calibration of sensors to be used for low-ambient sampling). 9. Abstract has some awkward

statements that could be improved, as well as providing more quantitative results. e.g., "live, work, breathe. . ." – breathing is something that happens at all locations. . .one would hope. Also what is meant by "stakeholders"? The public? Industry? 10. Did the authors ever characterize the response time of the sensors? (e.g., against high time-resolution instruments also made by Aerodyne). A brief statement on their utility for a mobile sampling approach and time base of the data would be helpful, as many low cost sensor systems are being employed in a mobile fashion.

---

## Short Comment (SC1) · 7 Jul 2017

**Comments on Cross et al. (2017) from N. Zimmerman, R. Subramanian, A. Presto and A. Robinson, CMU**

**General Comments**

This paper discusses using HDMR to calibrate the low-cost sensors used in the Aerodyne ARISense air quality monitor. While the results seem promising, it is difficult to assess the performance of the model, because the training data appear to have been included as part of the model performance assessment. This would bias the model performance and makes it difficult to compare the results with other studies that evaluate model performance using independent datasets.

Additionally, we believe the paper would benefit from more discussion on building and interpreting the HDMR model. Questions such as what was the maximum order used, what variables were significant, and any physical interpretation of any significant variables are either missing or underdeveloped. The paper would also benefit from some additional metrics of model performance beyond correlation plots.

Another question to address is how the training data are chosen. From Figure 6, it appears that only periods where there were pollutant concentrations were elevated were chosen to build the model. How could this calibration approach be generalized for others? If the training data set was carefully constructed vs. randomly selected then is it feasible to assume that the model training window could be condensed to 1 week, as the other reviewers suggest?

As a full disclosure, we are also in the process of submitting a manuscript on a different type of calibration model for low-cost electrochemical sensors. We welcome and encourage feedback from Aerodyne on our manuscript in kind to help the community collectively improve sensor performance.

**Specific Comments**

Page 5: Line 26-27: Can you be more specific? What is your definition of "acceptable accuracy" –the paper would benefit greatly from some quantitative performance metrics.

Page 6, Line 11-12: What is the statistical analysis done to decide which variables are significant? Something like AIC/BIC? ANOVA? T-test?

Page 6, Line 13-14: I am not sure I fully understand the HDMR. Can the orthogonal basis functions be written in closed form (parametric?) I think a couple extra sentences here introducing the model are warranted.

Page 6: Line 20-23: What is the spanned range? For others building their own co-location windows, what were the critical criteria to determine the optimal co-location period? Was 35% arbitrarily chosen or was the calibration window tuned and if so, what was learned during tuning? Some discussion of diminishing returns vs. training window would be helpful to others implementing these methods.

Page 6 Line 12-18: This is another paragraph where I think some quantitative performance criteria would be useful. When comparing the performance of HDMR calibrations to manufacturer corrections or corrections by other papers, it's not clear what the terms 'reasonably good correlation' or 'relatively small' mean.

Page 7 Line 26-27: It seems like a lot of interesting work was done in the lab, but none of these results are provided. I'd be interested to see more details here. Can this be included in supplemental?

Page 8, Line 14: What was the environmental variability spanned? And how was the 35% subset chosen? This is a follow up to the previous comment.

Page 8, Line 20: This seems problematic, was the performance of the model tested on a data set in which 35% was used for training? Ideally the model should be tested on completely blind test data (i.e., the remaining 65%). If this is what you did, it should be made clearer. If this is not what you did, you should provide performance metrics for the pure testing data since this approach is the only way to truly test the model performance.

---

## Author Comment (AC1)

**Use of electrochemical sensors for measurement of air pollution: correcting interference response and validating measurements, Cross et al., submitted to AMTD 5 May, 2017.**

We thank the reviewers for their careful reading of the manuscript and many helpful comments. Our responses to specific comments can be found below. The reviewers' comments are in italics, our responses are in plain font, and changes made to the manuscript are in quotation marks and indented. Page and line numbers refer to the original document. We have made significant changes to some of the figures and have included new performance metrics for the HDMR model. However, none of these changes affect the conclusions in the manuscript.

**Response to Reviewer RC1: Anonymous Referee #1**

*This is a rigorous attempt to illustrate the challenges and utility of deploying 'low cost' air quality sensors in a community. This is a field of growing interest – likely to become more crowded – with important implications across most spheres of atmospheric research. A particularly strength of this work is stressed by the authors in warranting caution in interpretation of data from these types of sensors.*

*Specific comments: P1, Line 14: perhaps this is better phrased as '…address environmental justice issues related to air quality.' P1, Line 22: 'Protecting the air environment…' isn't really one of the most important PH challenges. Rather, it is protecting populations from degraded air quality exposure that is important. P1, Line 29: to presume the authors mean US dollars? P8, Line 18: extra word 'in'*

We have made the suggested changes to wording.

> P1, line 14: "…address environmental justice issues related to air quality."

> P1, line 22: "Protecting populations from exposure to poor air quality is…"

> P1, line 29: "…thousands of US dollars."

> P8, line 18: deleted "in"

*General Comments: One issue that is not discussed is the potential for lot variability in sensor performance within a single manufacturer. Whilst the authors provide adequate detail on which make/model has been chosen (P2, L33-40), do these EC sensors exhibit differences within a manufacturer production lot? Or are there differences across different lots?*

This is an important question, and one that lies at the heart of the low-cost sensor calibration challenge. Based on our empirical experience, we have observed significant (up to a factor of 2.5x) differences in sensitivity for batches of otherwise identical sensors. In these cases, the sensors themselves have the same age (relative to manufacturer production and out-of-package timescales). This evidence supports the need to build sensor-specific (and in the context of multi-pollutant measurement systems, system-specific) calibrations as opposed to a general network-wide calibration. Based on our limited observations, the manufacturing process for these electrochemical sensors is not yet fully reproducible.

We have revised the end of Section 2.1 (page 3) to address this issue:

> "This paper presents results for the four electrochemical sensors in a single ARISense system. Note that nominally identical electrochemical sensors can have widely different sensitivities and

exhibit variable environmental interference effects. As a result, the specific calibration models described in this paper cannot be broadly applied to all ARISense systems. Until the reproducibility of electrochemical sensor manufacturing improves, system-specific HDMR models will need to be developed for each individual ARISense system to maintain sensor quantification metrics."

*The paper begins with a discussion on environmental justice (abstract), and includes very specific references to asthma rates in the sampled community (P4, lines 1-4). Context is, of course, important, but these facts seem out of place in this manuscript which is mainly a focus on the technical details of using and interpreting EC sensors.*

We have removed some of the details about asthma rates. The end of the first paragraph on page 4 now reads:

"The original DAQSS deployment and initial ARISense proof-of-concept efforts were motivated by the need to assess the viability of lower-cost AQ sensor systems in communities suffering from environmental health knowledge gaps, such as the unexplained doubling of the adult asthma rate in North Dorchester between 2001 and 2010 (Backus, 2012)."

*It was surprising to see a reported temperature range of 5-45 degrees in a northern US city, but the authors later state that this was internal box temperature to assess electrode function under actual operating conditions. When comparing these data, was a correction to ambient temp and RH taken into account (e.g. temp/RH measured by a nearby met station)? For example, if ambient temp and RH were 25 deg and 50%, but the internal temp where 35 deg and 15% because of strong sunlight, one would expect a significant effect, given the apparent sensitivity.*

Due to the sensitivity of the electrochemical sensors, it is more important to consider measurements of temperature and RH at the sampling interface of the sensors rather than ambient conditions, as these internal sensor-specific conditions will most closely correspond to the interference signal observed by the working and auxiliary electrodes. However, comparison of sensor system temperature and RH inside the flow-cell with ambient temperature and RH showed that even under conditions of direct sunlight, the sensor system internal conditions remained within 10-15% of the ambient values throughout the co-location sampling period. We have expanded our explanation of the temperature and RH measurements in the 4th paragraph of section 2.1 as follows:

"The manifold includes an embedded RH/T sensor positioned adjacent to the electrochemical cells which is used to model the temperature and relative humidity-derived interference effects on the raw sensor response. Given the active flow of the gas sampling inlet and minimal residence time (~1s) of the sample air within the manifold, the RH and T measurements recorded by the ARISense system closely track changes in ambient RH and T conditions. Over the co-location period described here, measurements inside the flow manifold were within 10-15% of the ambient values even under conditions of direct sunlight."

*Why was there no data included or discussed for particulate matter or CO2?*

We explained at the end of the third paragraph on page 3 that the particulate matter measurements will be assessed in a future manuscript. Due to the size detection limit for the OPC, and the anticipated size distribution of near-field and accumulation mode aerosol particles in most urban environments – the utility and reliability of low-cost OPCs for PM2.5 measurements remains highly uncertain. As such, significant analysis effort is required to reconcile OPC metrics and this is the subject of ongoing work in

our laboratory. Since this paper focuses on calibration of electrochemical sensors, we chose not to include the $CO_2$ data. We have added a sentence at the end of the second paragraph on page 3 explaining this:

> "Note that the $CO_2$ measurements are not discussed in this paper which focuses on the electrochemical sensors, but will be addressed in a future manuscript."

*In a number of cases, the authors refer to this sensor package as a 'low-cost' replacement for measuring air quality, which could play a key role towards empowering environmental justice (P9, Line 34). The authors are correct in asserting that lower cost sensors likely have a role in improving granularity in air quality monitoring networks, especially in locations with disproportionate air quality burdens, like the relatively low income communities in which this study takes place. But the idea of 'low cost' is a fairly subjective statement that seems meant to broaden the appeal of these products to communities in need. The development of low or lower cost sensor units with an eye towards reducing injustices is a noble and important direction for air quality scientists, but it might provide value to compare this instrument against the few other existing low/lower cost sensing units that are found in the literature – both in terms of sensor performance and relative cost.*

The reviewer makes an excellent point that "low-cost" is a subjective term. We have changed the designation for ARISense throughout the paper from "low-cost" to "lower-cost," and have added the following sentences in the introduction to explain the rough cost tiers for air quality monitoring systems. It is difficult to provide the exact cost of other Tier 2 systems (integrated multi-pollutant measurement packages) since this information is often proprietary. We have also included a table summarizing recent published performance data for lower cost AQ systems that utilize Alphasense electrochemical sensors in Table 4.

> "Air quality monitoring systems can be roughly divided into three cost tiers, 1) high cost/high accuracy systems costing tens to hundreds of thousands of US dollars, such as those used at regulatory monitoring stations, 2) lower cost systems costing a few to ten thousand US dollars, such as the ARISense system or the recently developed Real-time Affordable Multi-Pollutant (RAMP) package developed by Carnegie Mellon University and Sensevere (Zimmerman et al., 2017), and 3) low cost systems (costing tens to hundreds of US dollars) designed for the consumer market that typically only measure a single pollutant and generally suffer from poor quality data (EPA, 2017). The goal of second tier systems is to provide data quality approaching Tier 1 at a fraction of the cost."

*The largest issue seems to be in interpretation and setup of the HDMR model to adjust sensor data to real values. Specifically, the authors state that the model can 'capture the intricate interdependencies of the variables. . .' in order to correct the data and provide guidance to researchers which variables are most impactful (P6, Line 1). These statements presume that the researchers enter in all possible variables that are likely to play a role in sensor performance. Given the relative few number of variables measured, and presumably computed, how can a researcher have confidence that they are accounting for all – or at least most – of the likely variants that may affect their results? The concern here is that there may be other plausible covariates that affect sensor performance. For example, one might imagine a measure of $CO_2$ by NDIR could be affected by water vapor (which is imputed by this sensor package), but also by other ambient IR-absorbing components (that are not measured)?*

To re-phrase the reviewer's question – they are asking about the challenge of unknown unknowns – if there is a specific interference vector that is not explicitly measured (and modeled) with the system, how can we be confident that the results from the integrated system will remain robust in the real-world

presence of this interference vector? To first order, we are confident that the model can handle the full extent of environmental interference vectors encountered in the Dorchester micro-environment based on the results of the test data which leverages the long-term co-location of the sensor with reference measurements. By combining on-board measurements of multiple pollutant species (via raw sensor outputs – both WE and Aux electrode signals) and environmental conditions (P, T, RH) and allowing the ambient variability of these species and conditions to train the model, the results suggest that the primary factors impacting sensor performance are captured by the model for this set of electrochemical sensors (although clearly, the Ox-B421 sensor is underperforming relative to the others). It is certainly possible that the NDIR measurement of $CO_2$ could be impacted by cross-sensitivity to species not measured by the other sensors in the system, but modeling and validating the NDIR sensor response is beyond the scope of the current manuscript and will be examined in a subsequent work.

*The authors also note (P 6, Line 8-9) that in the first step of modeling, a user can choose how many variables are selected to interact with one another. How does one quantitatively make this determination?*

This comment overlaps with comments from Reviewers 2 and 3, requesting more information on how the model was optimized. We have added the following paragraph on page 7 and to describe the development of the HDMR model for the NO sensor:

> "An example of how the HDMR model is developed for the NO-B4 sensor is provided in the Supplemental Material. The left column of Table S2 lists all available input parameters and the other columns denote which parameters were included in the input matrix for each model run. The bottom rows list the RMSE, MAE, and MBE for each model run for both the training data (model generation) and test data (model evaluation)."

We have added the following text, table and figure to the Supplemental Material:

> "Table S2 shows a subset of the input matrices for training the NO-B4 sensor output to the NO reference measurements. Six model versions are shown (labelled v.8-13), and the resultant RMSE, MAE, and MBE (in ppb) are listed at the bottom of the table for both the training set and the test data (with test metrics shown in curly brackets). The optimal model run (v.8) is indicated with shading. The table shows that while the model with the most diverse set of inputs (v.12) resulted in the lowest RMSE, MAE, and MBE values for the training data, its RMSE and MAE were worse compared to v.8 when applied to the ambient test data. It should be noted that *ExploreHD* also performs statistical F-tests to further refine which inputs and input pairs are considered in the HDMR model training, and to determine a suitable degree for polynomial basis functions for each component function.
>
> The poorer performance of model v.12, trained with the full set of inputs available can be explained by increased overfitting related to the additional degrees of freedom from the increased number of input pairs. The F-tests performed by *ExploreHD* during model generation are aimed at mitigating issues with overfitting, but only consider each input independently. Thus, this automated input selection is not perfect, especially for cases like electrochemical sensor quantification, where there is significant correlation between certain inputs in the training data. The approach used here of testing a range of input sets, effectively serves as a manual supplement to the automated input selection performed by *ExploreHD*.
>
> From Table S2 it is also seen that models excluding key inputs (*e.g.* v.10) exhibit poorer performance on test (and training) data. The input selection used in model v.8 exhibits a reasonable trade-off between the issues of exclusion of important inputs and overfitting (as the

performance on training and test datasets were comparable in this case). Through future work, it may be possible to refine or replace the F-test-based input selection algorithm used by *ExploreHD* so that overfitting might be addressed in a more automated fashion for training datasets exhibiting high correlation between certain inputs.

Figure S1 shows key input pairs in the NO HDMR v.8 model. The figure plots normalized total sensitivity indices for the input pairs. These sensitivity indices quantify the proportion of variance that can be explained by each input pair, considering both structural and correlative components. These metrics are the result of a structural and correlative sensitivity analysis (SCSA) performed by *ExploreHD*, which is described in [Li et al. "Global Sensitivity Analysis for Systems with Independent and/or Correlated Inputs", *J. Phys. Chem. A. 114*. **2010**, 6022]. In addition to calculating a total sensitivity index for each input / input pair, this analysis decomposes the total sensitivity into structural contributions reflecting the underlying system model, and correlative contributions reflecting covariation between inputs in the dataset being considered. Decomposition of sensitivities in this manner provides the opportunity for additional insights into the role of each input / input-pair."

**Table S2. Set of HDMR models for NO sensor. The optimal model (v.8) is indicated with shading.**

| Model v. | 8 | 9 | 10 | 11 | 12 | 13 |
|---|---|---|---|---|---|---|
| CO AUX | x | | | | x | x |
| CO WE | x | | | x | x | x |
| NO AUX | x | x | | | x | x |
| NO WE | x | x | x | x | x | x |
| NO2 AUX | | | | | x | x |
| NO2 WE | | | | | x | x |
| Ox AUX | | | | | x | |
| Ox WE | | | | | x | |
| Dew point | x | x | x | x | x | x |
| Temperature | x | x | x | x | x | x |
| CO2 (voltage) | | | | | x | x |
| RMSE (ppb) | 3.38 | 5.09 | 6.75 | 4.88 | 2.58 | 2.81 |
| { test } | { 4.52 } | { 5.86 } | { 7.16 } | { 5.53 } | { 9.19 } | { 6.41 } |
| MAE (ppb) | 2.40 | 3.29 | 4.23 | 3.05 | 1.63 | 1.76 |
| { test } | { 2.83 } | { 3.90 } | { 4.94 } | { 3.27 } | { 4.07 } | { 3.27 } |
| MBE (ppb) | 0.02 | 0.32 | -0.55 | 0.15 | -0.01 | -0.05 |
| { test } | { 0.97 } | { 1.80 } | { 2.08 } | { 1.08 } | { 0.30 } | { 0.87 } |

[Figure]

**"Figure S1.** Normalized total sensitivity indices of each significant (contribution > 0.1%) input pair in NO Model v.8. Of the possible combinations, the NO-WE/Temp, NO-WE/NO-AUX, and NO-AUX/CO-WE explain more than 80% of the sensor-system variance trained against the corresponding reference [NO] measurement."

*It is very difficult to discern useful results from Figure 3. Further, we must presume that these data have been validated by the investigators. If so, it is surprising to see spikes of ozone exceeding 1000ppb with some regularity in this location, as observed by the reference monitor.*

We agree with the reviewer that Figure 3 is not very useful. We have therefore removed it and included a 72-hour segment of the differential voltages in what was formerly Figure 5 (now Figure 3). The reference data for $O_3$ plotted in the original Figure 3d was incorrect (displaying reference CO data instead of $O_3$ data in the AMTD version of the paper). We apologize for this error.

*Figure 5 is a fairly useful illustrative figure that clearly identifies sensor limitations. But it is troubling to see divergence between the EC sensor and the reference sensor in periods of relative stability in temperature. This seems to need further explanation – how does your data compare for this specific time series after it has been modeled?*

Temperature is not the only variable driving the divergence between the EC and reference sensors, hence the need for a complex, multi-dimensional model to explain the variance in the raw sensor relative to reference. We have revised Figure 5 (now Figure 3) to include the model output and we have revised the text on page 7 (now page 8) as follows:

> "Figure 3 shows the time-series for a ~ 72-hour period for the relative humidity, dew point temperature (panel a, solid and dashed lines, respectively), temperature (grey shaded area), and raw differential sensor output (dashed line), reference measurement (thick red dashed line) and model output (thin solid line) for the four electrochemical sensors (panels b-e). The raw differential sensor output is displayed as a voltage ($\Delta mV$) which is linearly proportional to the difference in current generated within the electrochemical cell at each electrode (working and auxiliary). The correlation plots between the raw EC-sensor output and the reference measurements are shown in Figs. 4a to d, with each data point colored by flow-cell temperature. The intercept, slope and $r^2$ for the linear regression indicated with a red line in Fig. 4 are listed in Table 1."

with additional discussion on page 10:

> "Closer examination of the model output for 72-hours of the test data in Fig. 3 gives additional clues for improving the model. In Fig. 3d at ~18:00 on 11/2/2016, the model $NO_2$ exceeds the reference $NO_2$ by a factor of ~2 during a period of rapidly decreasing temperature and increasing RH. This underscores that the rate of change of input parameters may be important in the model, in addition to the absolute values. Fig. 3e also suggests that the HDMR model for $O_x$ struggles during times of rapidly changing temperature, particularly when the $O_3$ concentration is low (< 3 ppb). Future development of HDMR models to support ARISense quantification will include derivatives of key variables as inputs."

*The authors included a number of variables to consider in adjusting or training the model, but specifically excluded sensor age, noting that the sensors were approximately 6-7 months old at the end of the study and, therefore, should have limited effect on model performance. Firstly, wouldn't it be more appropriate to compare sensor age to manufacturing date, rather than when a package is opened?*

Based on communications with the manufacturer of the sensors, sensor aging is directly related to the loss of electrolyte (7M $H_2SO_4$) from a given EC cell, which is primarily driven by exposure to extremes in RH (< 15% or > 80%). Under these conditions the electrolyte will either evaporate (<15%) from the cell or absorb significant $H_2O$ (> 80%), overflowing the cell. Upon completion of a batch of sensors, the

manufacturer ships the cells in self-contained sealed containers at 25C and 60% RH, following equilibration in their laboratory.  Therefore, we interpret the onset of sensor-aging (sensor t0) as the date at which this seal is broken for a given individual sensor.

*And second, it is unsatisfying to ignore sensor age as a relevant variable, given the relatively short lifetimes of these sensors. For example, the NDIR lamp and electrode has a lifetime of 2000-6000 hours (according to the manufacturer), depending on lamp light time and the presence of heavy contaminating pollution. This is 80- 250 days, which is not much longer than the study length presented here, and suggests that long term drive is something that should not be ignored.*

While we agree that modeling the decay (i.e., aging) of electrochemical sensors is extremely important, such an exploration is beyond the scope of the analysis presented here. The manufacturer quotes the following operating lifetimes (degradation of signal to 50%): 36 months for CO, and 24 months for NO, $NO_2$ and Ox. These timescales are longer than the 4.5 month deployment (6.5 mo. out of package) pertaining to the current work, and we therefore do not expect significant (>5%) sensor degradation due to aging.  A detailed assessment of sensor performance over 18-24 months of continuous ambient operation is ongoing and it is with this subsequent dataset that time will be considered as an input to the model in an effort to track and correct for degradation in performance over sensor lifetime.  The NDIR sensor is not discussed in this manuscript. We have revised the last paragraph of section 2 (page 8) to read:

> "The data presented in this paper were recorded over a 4.5-month sampling interval (July 7, 2016- November 23, 2016).  All four electrochemical sensors used in this study were first removed from their packaging on May 9, 2016.  That means that from out-of-package, the sensors had aged ~6.5 months by November 23.  The manufacturer quoted lifetime for degradation of the signal to 50% is 36 months for the CO sensor and 24 months for the NO, $NO_2$ and $O_x$ sensors.  Given that these lifetimes are significantly longer than the deployment time scale analysed here, we did not include a time-dependent sensitivity term in the input matrix of our HDMR model runs.  The results presented here therefore assume that the sensitivity of each of the electrochemical sensors did not appreciably drift over the 4.5 month deployment.  In subsequent studies we will analyze sensor response over longer deployment timescales (18 to 24 mo.) to investigate the importance of including a time-based parameter to track and correct for drift in sensor response with time."

**Response to Reviewer RC2: Anonymous Referee #2**

*General comments: This paper is timely in describing how to improve the performance of a set of Alphasense electrochemical sensors, which are being widely incorporated into may emerging multipollutant air quality sensor technologies. The paper goes into great depth in exploring causes of sensor measurement artifacts and demonstrates an approach to improve the data quality. However, this paper will have a limited impact if several important issues are not addressed. A recommendation of major changes is suggested, focusing upon these areas of improvement:*

*1. How are authors defining "good enough" for sensor data quality? They indicate a goal of having credible data and "acceptable accuracy" (line 27), but need to clarify what they consider to be their target (accuracy, measurement range, etc.) and for what purpose.*

We thank the reviewer for this important question and have revised the paper to include the performance metrics of root mean square error (RMSE), mean absolute error (MAE) and mean bias error (MBE). We have added the following paragraph on page 7 and the following table to the Supplemental Material.

> "The metrics used to evaluate the model are the slope and intercept of a linear least squares regression of the model output with the reference measurements, the coefficient of determination of the linear fit ($r^2$), the root mean square error (RMSE), the mean absolute error (MAE), and the mean bias error (MBE). The equations for these metrics are given in Table S1 and model-to-measurement results are summarized in Tables 2, 3, and 4."

**Table S1. Metrics used for comparing EC sensor model output ($y_i$) to reference measurements ($x_i$).**

| Statistic | Abbrev. | Formula | Description |
|---|---|---|---|
| Coefficient of determination | $r^2$ | $r^2 = 1 - \dfrac{\sum_{i=1}^{n}(y_i - f_i)^2}{\sum_{i=1}^{n}(y_i - \bar{y})^2}$ | $f_i$ is the value of the linear least squares fit at $x_i$. Ratio of explained variation to total variation. For linear least squares regression, $r$ is equal to Pearson's correlation coefficient. |
| Root mean square error | RMSE | $RMSE = \sqrt{\dfrac{1}{n}\sum_{i=1}^{n}(y_i - x_i)^2}$ | Standard deviation of difference between model output and reference values. Measure of accuracy. Sensitive to outliers. |
| Mean absolute error | MAE | $MAE = \dfrac{1}{n}\sum_{i=1}^{n}|y_i - x_i|$ | Average of the absolute error. Disregards the direction of under- or over-prediction. |
| Mean bias error | MBE | $MBE = \dfrac{1}{n}\sum_{i=1}^{n}(y_i - x_i)$ | Average error. Indicates if model output values are biased high or low relative to reference values. |

*2. The authors note in their concluding sentence that "This compression of the training period is especially important. . ." Currently, they used 35% of a 4 month period of data to develop a complex model to improve the data. Why 35%? What is the performance if only 10% of the data were used? What if only the first week of data were used? Authors have sufficient data to explore the implications of different training periods that would provide important insight to researchers looking to employ sensors and develop study plans yielding reasonable data quality. It is recommended that authors go into substantially more depth to investigate the training period required.*

The goal of this work is to show that with a sufficiently wide range of input parameters, we can successfully use the HDMR model to analyze electrochemical sensor raw output data. We did not systematically attempt to minimize the training period. Instead, we focused this initial modeling effort on evaluating whether the HDMR model could yield robust results when trained with a fairly conservative (intentionally comprehensive) set of input data (incl. range of gas concentrations, magnitude of environmental conditions and rates of change of environmental conditions). However, in our re-analysis of the test and training data points prompted by this question, we realized that we had unnecessarily eliminated test data by requiring that all four reference measurements be sampling ambient simultaneously (with none of the reference instruments in auto-calibration mode). Because the reference instruments all have different calibration schedules, this inadvertently decreased the set of test data. By treating each individual reference measurement separately, we were able to recover ~ 10% of the total test data and use that to evaluate the HDMR models. As a result, the training data now constitutes ~ 24-27%

of the total eligible co-location data points, leaving ~73-76% of the data available to test the HDMR model against ambient pollutant/condition variability.

We have also added a figure to the supplemental material to show the distribution of parameters used during the training and test periods. And we have added the following text on page 9 to clarify our goals:

> "The training data for the HDMR model were chosen to provide comprehensive coverage of environmental variability spanning the July-November sampling interval. It was important to include (1) sensor responses to the range of gas concentrations encountered in ambient air (near-zero to high concentration transient spikes in pollution), (2) the range of temperatures and various rates-of-change in temperature, and (3) the range of measured water content of the sample air in the flow-cell. The goal was to include a wide enough range of training data to avoid extrapolation errors when applying the model to the test dataset (all ambient co-location data not included in the training dataset). Figure S3 shows the distributions of temperature, reference measurement, dew point temperature and relative humidity for the training data for the CO-B4 HDMR model, overlaid with corresponding distributions of the test data. We did not attempt to minimize the amount of ambient data used for training, or vary the timing of the training data with respect to the test data. Approximately 25% of the full time series was used to generate the model (Table 2 and indicated with grey bars in Fig. 5). The exact fraction of data used for training was slightly different for each sensor due to differing calibration schedules for the reference measurements (which automatically excludes sensor data from the training or test datasets). For each sensor, the set of inputs included in the input data matrix was optimized as described in Section 2.5 and the Supplemental Materials."

[Figure]

**Figure S3.** Distributions of temperature, reference measurement, dew point temperature, and relative humidity for the training data for the CO HDMR model (dashed lines/shaded) and the test (solid lines) data. In the case of CO, the training data distributions were generated from 27% of the total available co-located interval with 7974 5-min average data points comprising the model training matrix and 21533 5-min average data points comprising the test data.

*3. Authors should investigate an aging effect – they indicate they will only explore this later, but should at minimum demonstrate whether there is any relationship with the number of "out of box" or "in use" days. In Jiao et al (2016, https://doi.org/10.5194/amt-9-5281- 2016), aging was clearly demonstrated in a number of sensor types that incorporate Alphasense sensors.*

As noted in the response to Reviewer 1, the deployment time (4.5 months) was much shorter than the manufacturer quoted lifetimes (24 to 36 months) of the sensors. A detailed assessment of sensor performance over 18-24 months of continuous ambient operation is ongoing and it is with this subsequent dataset that sensor age will be considered as an input to the HDMR model formulation in an effort to track and correct for degradation in performance over sensor lifetime.

*4. How variable is the performance between identical sensors? How variable are the HDMR models from one RAMP to another?*

This work focused on results obtained from a single ARISense system, which is different from the Real-time Affordable Multi-Pollutant (RAMP) sensor package developed by Sensevere and Carnegie Mellon

University for which multiple units were evaluated in Zimmerman et al., 2017. As mentioned in response to Reviewer 1, there is significant variability at the manufacturing level of Alphasense electrochemical sensors, which results in the need to build sensor-specific calibration models across the same type (e.g., CO-B4) of sensor. Moving forward we anticipate that unique HDMR models will be necessary for each ARISense system, due to the irreproducibility of the manufacturing process as it relates to each electrochemical sensor. The aim of the current work is to provide the first demonstration of the HDMR model trained on an ambient co-location dataset. Future laboratory-based efforts will focus on building HDMR models for multiple nodes from a compressed (~1 week) training interval during which pollutant concentrations and environmental conditions are systematically varied to sample the full range of conditions relevant to the field.

We have revised the end of Section 2.1 (page 3) to clarify this issue:

> "This paper presents results for the four electrochemical sensors in a single ARISense system. Note that nominally identical electrochemical sensors can have widely different sensitivities and exhibit variable environmental interference effects. As a result, the specific calibration models described in this paper cannot be broadly applied to all ARISense systems. Until the reproducibility of electrochemical sensor manufacturing improves, system-specific HDMR models will need to be developed for each individual ARISense system to maintain sensor quantification metrics."

*5. The HDMR analysis is fairly opaque – authors cite papers that describe the approach, but do not provide sufficient detail for this to be reproducible. It is recommended that authors provide more specific information on the HDMR analysis and resulting model in the supplemental information. Given some sensor applications involve real-time transmission and display of data to the public, does the HDMR approach support this or must it be performed post hoc?*

As discussed above in the response to Reviewer 1, we have added a description of how the HDMR model is developed in the main text with supporting tables in the Supplemental Material. Real-time implementation of the HDMR model is feasible by adding the sensor-specific and system-specific models (in closed form, algebraic expressions) to the backend database architecture to enable real-time reporting of calculated ppb values.

*Minor comments:*

*- Quality of the text on figures needs improvement – recommend not using red font text and ensuring clear, readable axes.*

The red font text within figures has been modified and the font for axis labels has been increased in size and bolded.

*Authors compare against DEP monitors – they should indicate what are the detection limits of the monitors and implications for their calibration. Since regulatory monitoring stations are employed to evaluate air quality relative to the NAAQS, detection limits can be an issue in low concentration areas (e.g., some CO monitors have ~300 ppb detection limits, which may be fine for the NAAQS at a ppm level but may be an issue for co-location and calibration of sensors to be used for low-ambient sampling).*

We have modified section 2.3 to include the limit of detection (LOD) and RMS noise for each of the reference monitors used in the current study.

"The reference measurements used in this study include ozone ($O_3$, Teledyne Model T400 Photometric Ozone Analyzer, LOD <0.6 ppb; RMS < 0.3 ppb), carbon monoxide (CO, Teledyne Model 300EU Carbon Monoxide Analyzer, LOD < 20 ppb; RMS ≤ 10 ppb), and nitrogen oxides (NO, $NO_x$, $NO_2$, Teledyne Model T200 Nitrogen Oxide Analyzer, LOD = 0.4 ppb; RMS < 0.2 ppb)."

*Abstract has some awkward statements that could be improved, as well as providing more quantitative results. e.g., "live, work, breathe. . ." – breathing is something that happens at all locations. . .one would hope. Also what is meant by "stakeholders"? The public? Industry?*

We have deleted the word "breathe" in the first sentence of the abstract. We have replaced "stakeholders" with "public."

*Did the authors ever characterize the response time of the sensors? (e.g., against high time-resolution instruments also made by Aerodyne). A brief statement on their utility for a mobile sampling approach and time base of the data would be helpful, as many low cost sensor systems are being employed in a mobile fashion.*

ARISense v1.0 was designed to serve as a stationary, fixed site AQ node. Given that the fastest response times we could access from the DEP monitoring station equipment in this work was 60s averages, the time-response of the sensors themselves did not lag the rates of change in pollutant concentrations that were characterized by the reference instrumentation at the site. Laboratory studies are underway that will examine the response time of the electrochemical sensors more carefully and evaluate the limits of reconciling pollutant gradients from mobile measurement platforms (bicycle and drone), recording sensor metrics at 1Hz. This work will be the subject of a forthcoming publication.

**Response to Reviewer SC1: N. Zimmerman, R. Subramanian, A. Presto and A. Robinson**

*This paper discusses using HDMR to calibrate the low-cost sensors used in the Aerodyne ARISense air quality monitor. While the results seem promising, it is difficult to assess the performance of the model, because the training data appear to have been included as part of the model performance assessment. This would bias the model performance and makes it difficult to compare the results with other studies that evaluate model performance using independent datasets.*

We thank the reviewers for bringing this issue to our attention. This was a serious oversight on our part and we have now corrected it. We have now re-evaluated the model performance using the test data only (i.e., excluding the training data). The results are presented in Figures 3, 4 and 5 and the performance metrics are in Table 3. Excluding the training data from the test data decreased the $r^2$ for each sensor by approximately 10%, but did not change the conclusions of the paper.

*Additionally, we believe the paper would benefit from more discussion on building and interpreting the HDMR model. Questions such as what was the maximum order used, what variables were significant, and any physical interpretation of any significant variables are either missing or underdeveloped.*

As discussed in the response to Reviewer 1, we have expanded our discussion of building and interpreting the HDMR model in the main text and in the Supplemental Material.

*The paper would also benefit from some additional metrics of model performance beyond correlation plots.*

We agree with the reviewers that additional performance metrics besides the slope of the correlation plot and $r^2$ are important to include. As noted in the response to Reviewer 1, we have included RMSE, MAE and MBE as additional performance metrics.

*Another question to address is how the training data are chosen. From Figure 6, it appears that only periods where there were pollutant concentrations were elevated were chosen to build the model. How could this calibration approach be generalized for others? If the training data set was carefully constructed vs. randomly selected then is it feasible to assume that the model training window could be condensed to 1 week, as the other reviewers suggest?*

As noted above in the response to Reviewer 2, we chose training data that covered the full range of the parameters that would be encountered in the test data. We have added additional discussion in the main text and Figure S3 with the distributions of parameters measured during the training and test data periods.

*As a full disclosure, we are also in the process of submitting a manuscript on a different type of calibration model for low-cost electrochemical sensors. We welcome and encourage feedback from Aerodyne on our manuscript in kind to help the community collectively improve sensor performance.*

We look forward to commenting on your manuscript.

*Specific Comments*

*Page 5: Line 26-27: Can you be more specific? What is your definition of "acceptable accuracy" –the paper would benefit greatly from some quantitative performance metrics.*

Quantitative performance metrics are now included for the training and test datasets, listed in Tables 2 and 3 of the main text.

*Page 6, Line 11-12: What is the statistical analysis done to decide which variables are significant? Something like AIC/BIC? ANOVA? T-test?*

HDMR uses F-test as an initial evaluation of the relative importance of individual input parameters to a given trained output vector. As noted above in the response to Reviewer 1, a more thorough description of our approach to HDMR is now included in the Supplemental Materials.

*Page 6, Line 13-14: I am not sure I fully understand the HDMR. Can the orthogonal basis functions be written in closed form (parametric?) I think a couple extra sentences here introducing the model are warranted.*

Yes, the 2$^{nd}$ order polynomial fits (cubic as max) can be exported from the HDMR log files into a closed form, algebraic equation which in turn can be embedded in the backend database architecture of the ARISense server to provide real-time concentration metrics through the online user-interface. These equations are also embedded in the firmware for each ARISense system so that concentration values can be logged to the local on-board USB drive if the system is run in off-line mode.

*Page 6: Line 20-23: What is the spanned range? For others building their own co-location windows, what were the critical criteria to determine the optimal co-location period? Was 35% arbitrarily chosen or was the calibration window tuned and if so, what was learned during tuning? Some discussion of diminishing returns vs. training window would be helpful to others implementing these methods.*

As noted above in the response to Reviewer 2, we did not attempt to minimize or tune the training window. Our goal was to choose a set of training data that covered the full range of the input parameters that would be encountered during the test data.

*Page 6 Line 12-18: This is another paragraph where I think some quantitative performance criteria would be useful. When comparing the performance of HDMR calibrations to manufacturer corrections or corrections by other papers, it's not clear what the terms 'reasonably good correlation' or 'relatively small' mean.*

We have added a table to the main text (Table 4) summarizing results from three recent studies examining Alphasense electrochemical sensor-derived concentrations, tested against co-located reference measurements in ambient urban and suburban micro-environments.

*Page 7 Line 26-27: It seems like a lot of interesting work was done in the lab, but none of these results are provided. I'd be interested to see more details here. Can this be included in supplemental?*

While we agree with the reviewer that, in many ways, the laboratory setting provides a controlled environment across which the sensor response to a matrix of conditions can be characterized, our laboratory work with the ARISense system is ongoing and will be the subject of a forthcoming manuscript. The aim of that work will be to demonstrate that compressed (~1 week) training datasets can be generated through systematic laboratory experiments and that the resultant HDMR models provide a robust approach to ambient pollutant measurements made with the ARISense system across a variety of relevant micro-environments.

*Page 8, Line 14: What was the environmental variability spanned? And how was the 35% subset chosen? This is a follow up to the previous comment.*

These questions are answered above.

*Page 8, Line 20: This seems problematic, was the performance of the model tested on a data set in which 35% was used for training? Ideally the model should be tested on completely blind test data (i.e., the remaining 65%). If this is what you did, it should be made clearer. If this is not what you did, you should provide performance metrics for the pure testing data since this approach is the only way to truly test the model performance.*

We agree with the reviewers that including the training data in the testing data was problematic and regret the error.